# Quality of Life and Symptoms of PTSD during the COVID-19 Lockdown in Italy

**DOI:** 10.3390/ijerph18084385

**Published:** 2021-04-20

**Authors:** Sabrina Bonichini, Marta Tremolada

**Affiliations:** Department of Development and Social Psychology, University of Padua, 35131 Padua, Italy; marta.tremolada@unipd.it

**Keywords:** quality of life, COVID-19, Italy, quarantine, risk and protective factors

## Abstract

The COVID-19 pandemic has led to a complete lockdown in many countries and Italy was the first country interested in Europe, as the cases spread very quickly with a high rate of mortality. While the lockdown strategy is an essential step to curb the exponential rise of COVID-19 cases, it can have a significative impact on mental health on the population involved, that is still not well known and must be explored. The objective of the present research is to investigate the Quality of Life (QoL) and Symptoms of PTSD (PTSS) encountered during the quarantine period (April 2020) due to the spread of COVID-19 in Italy. Participants (N = 1839; 1430 females and 409 males), who were volunteers and anonymous, adults (18–73 years), were drawn from a convenience sample of the general population and asked to fill out an online questionnaire, after giving an informed written consent. The General Health Questionnaire (GH12), used to assess health related QoL, identified 24.5% of respondents as problematic, and the Impact of Event Scale-Revised (IES-R), used to assess PTSS, identified the 23.5% with clinical scores. Results showed that married people/cohabitants, non-workers, and those with a lower level of education perceived a better QoL and less PTSS. The most frequent emotions felt during the first month of quarantine and referred to by participants were sadness (72%), boredom (54.5%), impotence (52%), and anxiety (50%). The COVID-19 pandemic represents one of the most stressful events in recent times worldwide and poses a major challenge for social, economic, and, above all, psychological resources of the population that must be assessed and supported if insufficient.

## 1. Introduction

COVID-19, or Corona Virus Disease 19, the acute respiratory disease caused by the SARS-CoV-2 virus, was officially identified on 31 December 2019 in the city of Wuhan, the capital of the Hubei province, in China. Between 21 and 22 February the first infections were recorded in Italy, located in some towns of the Lodi area in Lombardy and in the Paduan area of Vo’ Euganeo (Veneto, Italy). In response to these events, the first Italian decree was passed to counter the effects of the coronavirus in the hope of being able to keep the spread of the disease under control. At the beginning of March, the infections in Italy rose to more than 3000 cases, leading to the enactment of a new provision that sanctioned the closure of the main meeting places, including schools, universities, cinemas, theatres, and stadiums, and the recommendation to implement in all possible cases smart working and to drastically reduce all social contacts, keeping a safety distance of at least 1 m. As March advanced, cases continued to grow exponentially, and so on 9 March, with a new decree, Italy became a “red zone”, and a lockdown was proclaimed. On 11 March 2020, the WHO declared that a pandemic was in effect.

During the COVID-19 pandemic, it has been imperative to understand how the populations, especially those in the severely affected countries, such as Italy, have been coping with such a major event. The aftermath that quarantining can have on a psychological level largely depends on the personal characteristics of each individual, based on personal history and distinctive elements. Apart from individual differences, there are also some general consequences that can be found in most people who have gone through a period of quarantine.

### 1.1. Quality of Life and Symptoms of PTSD during a Quarantine

Research conducted during previous epidemics (such as SARS) has documented a wide range of psychosocial effects on both individual and collective levels [1]. In a review published in February 2020 by Brooks et al. [2], a series of studies related to the psychological impact that a quarantine had on individuals during the SARS epidemic were analyzed. Most of them reported as possible consequences: PTSD symptoms, confusion, and anger. The vast majority of people who are forced to quarantine perceive this experience as fundamentally negative. Being distant from one’s family, boredom, and the partial loss of one’s freedom have potentially harmful effects on psychological balance. Risk factors such as being female and socio-economic low level were associated with a higher risk of developing SARS-related post-traumatic stress symptoms [3]. Other predictive factors mostly associated with a worse psychological impact during a quarantine that emerged from the pre-COVID-19 literature were: Young age (16–24 years), less schooling, and having only one child [4]. In addition, having a previous psychiatric illness [5] and working in a health profession [6] expose people to a greater risk of developing psychological problems such as post-traumatic symptoms, anger, fear, frustration, guilt, helplessness, isolation, loneliness, nervousness, sadness, and worry.

During a quarantine, in addition to socio-demographic factors, the factors that seem to have a worse psychological impact are as follows: Duration of the quarantine [6,7]; fear of being infected and being able to infect others [8,9]; boredom and frustration caused by the loss of daily routines and the reduction of physical and social contacts [6,10]; lack of basic necessities (food, water, clothing) [5,11]; and little or inadequate information [10,12].

To date, there already are studies that have examined the severity of the impact of the COVID-19 quarantine on the quality of life and symptoms of PTSD. Zhang and Feei Ma [13] investigated the impact of COVID-19 in Jinzhou, 2 weeks after the start of the lockdown in their country (China) on 263 participants (106 males, >18 years) using the Impact of Event Scale (IES). The overall scores on the IES indicated a slight impact of the lockdown on the QoL and symptoms of PTSD of their participants. In line with previous studies [14,15], well-being was lower for women than for men. Other recent studies on COVID-19 confirmed that being female conferred greater risks for depression, anxiety, and higher levels of stress [16,17,18,19,20,21].

To sum up, many studies have been already published documenting the negative psychological impact of a quarantine, but there is limited research that has investigated how severe the impact of the initial phases of the COVID-19 lockdown was on the quality of life and symptoms of PTSD. Therefore, the present study aimed to investigate whether there was an immediate impact of the COVID-19 lockdown on the quality of life and symptoms of PTSD among Italian adults, after 2 weeks from the beginning of the quarantine, when many restrictions were imposed by the Italian government.

### 1.2. Research Aims

The principal questions of this study were the following:How did Italian adults report their health related QoL and PTSS during April 2020 (initial phase of the lockdown), according to normative parameters?What were the emotions felt by the participants during this lockdown period?Which socio-demographic, daily life parameters, and emotions impacted on QoL and PTSS?

## 2. Materials and Methods

### 2.1. Participants

Participants (N = 1839) were all Italian-speaking adults (>18 years), with a mean age of 32.63 years (SD = 13.03, range 18–73), 1430 females and 409 males. Most of them were working (N = 1068), with a mean of smart-working/daily hours of 3.44 (SD = 3.43). Table 1 shows the socio-demographic characteristics of the sample.

### 2.2. Procedure

A cross-sectional study was performed from 1 to 30 April 2020. Only adults (aged > 18 years, no limit age) of Italian nationality, who provided a written informed consent (via Google moduli), were recruited in the study, using convenience and snowball sampling methods. The respondents’ participation was completely consensual, anonymous, and voluntary. The study protocol was approved by the Psychology Ethics Committee of the School of Psychology, University of Padua (number of protocol: D6B09283C9694D9C8EFCFBD33C713130).

Participants were asked to complete an online questionnaire via Google moduli requesting approximatively 20 min for filling it in.

### 2.3. Instruments

#### 2.3.1. Socio-Demographic and Daily Life Parameters Questionnaire

This part of the online questionnaire was used for the collection of information about socio-economic variables and cultural level of the participants, with questions investigating age, level of education (low = ISCED 0 to 2, primary and middle school; medium = ISCED 3 to 5, upper secondary school; high = ISCED 6 to 8, bachelor, master, PhD diploma), occupation status (working/not working), relationship status (married cohabitating, engaged not cohabitating, and single), number of smart-working hours and number of people living together during quarantine (Table 1). We also collected information about some daily life parameters: Presence of precedent pathologies, separation from people whom a participant used to live with before the quarantine, familiars directly involved in the emergency, economic difficulties caused by COVID-19, economic difficulties not caused by COVID-19, presence of close persons affected by COVID-19 with severe symptoms and familiars with a disability (Table 2).

#### 2.3.2. General Health Questionnaire (GH12)

Participants were asked to complete the General Health Questionnaire (GH12), a 12-item validated questionnaire assessing the quality of life. The GH12 [22,23] provides a reliable, valid, and brief assessment of QoL. The 12-item self-rated questionnaire measures physical health, psychological health, social relationships, and environment during the past 2 weeks. Each item is rated on a 4-point Likert scale (“less than usual”, “no more than usual”, “rather more than usual”, “much more than usual”), referring to the past 2 weeks, with a range of total scores from 0 to 36. Higher scores indicate a more problematic situation and there are different cut-offs to consider: Scores ≥ 15 are viewed as “problematic” and scores ≥ 19 as “very problematic” [3]. An example item is: “Able to concentrate”. On the whole, the higher total scores on the GHQ indicate higher psychological discomfort experienced by the respondents and vice versa. GH12 has good to excellent psychometric properties of reliability and performed well in preliminary tests of validity [22] that were also on the Italian population [24,25]. Cronbach’s alpha for this study was 0.80.

#### 2.3.3. Impact of Event Scale-Revised (IES-R)

The Impact of Event Scale-Revised (IES-R) [26] is a 22-item self-report measure that assesses subjective distress caused by traumatic events. Items correspond to 14 of the 17 DSM-IV symptoms of PTSD. Respondents are asked to identify a specific stressful life event (in this case COVID-19) and indicate how much they were distressed or bothered during the preceding 7 days by each difficulty listed. Items are rated on a 5-point Likert scale ranging from 0 (“not at all”) to 4 (“extremely”), with a lower score indicating a less stressful impact. The IES-R yields a total score (ranging from 0 to 88), subscale scores can also be calculated for Intrusion, Avoidance, and Hyperarousal. The IES-R is not used to diagnose PTSD; however, cut-off scores for a preliminary diagnosis of PTSD have been considered [26]. An example question is: “Any reminders brought back feelings about it.” A meta-analysis on IES confirmed its reliability, validity, and sensitivity as a measure of responses to stress in various populations [27] and in the Italian population, as well [28]. The Italian validation indicated the total score of 33 as a significative cut-off for PTSD presence.

Cronbach’s alpha for this study was 0.92 for the total score and ranged from 0.79 (Avoidance) to 0.87 (Intrusion) for the three subscales.

#### 2.3.4. Negative/Positive Emotions

Participants were also asked to indicate which emotions they were feeling during the last week from a list of 13 emotions both negative (i.e., pain, sadness, confusion, anger, boredom, disdain, shame, impotence, anxiety, sense of blame) and positive (i.e., interest, happiness, surprise).

### 2.4. Statistical Analysis Plan

Statistical analysis was performed using SPSS ver. 16.0 (IBM, Chicago, IL, USA). All results of quantitative variables were reported either as the mean and standard deviation or as frequency (percentage) (%). Pearson’s correlations were run to verify links between normally distributed variables. ANOVAs were employed to assess the difference in dependent variables, according to independent variables. As the effect size measure, η^2^ was used to categorize small (η^2^ = 0.01), medium (η^2^ = 0.06), and large (η^2^ = 0.14) effects for ANOVAs. Hierarchical regression models were run to identify the socio-demographic, daily routine and emotions factors impacting on GH12 scores as the dependent variables. Basic assumptions of linear regressions were checked computing Tolerance and VIF (for collinearity assumption), with the Durbin-Watson test (for assessing independent errors). A *p*-value < 0.05 was considered to be statistically significant.

## 3. Results

### 3.1. Daily Life Parameters

Before describing the results, descriptives of daily life parameters, considered in the following statistical analysis, are reported in Table 2. The majority of the respondents (from 74.4 to 97.1%) did not suffer from precedent pathologies, nor had to separate from people whom they used to live with before the quarantine, nor had economic problems, nor had a close person contract COVID-19 with severe symptoms. Only a few (14.6%) lived with familiars with a disability.

### 3.2. Quality of Life (GH12)

The overall mean GH12 score in participants amounted to 17.86, with an SD of 5.85 (within a range from 0 to 36), reflecting a contingent moderate stressful impact on QoL.

The General Health Questionnaire identified 39% of respondents as having subclinical QoL scores (score ≥ 15), 24.5% of such respondents as having very problematic scores (score ≥ 19), and 36.5% of such respondents as having normal scores (score < 15). GH12 scores were normally distributed, so parametric analyses were considered to assess the differences in the dependent variable according to sociodemographic.

ANOVAs showed there was a significative difference (F_(2, 1.836)_ = 5.50, *p* = 0.004, η^2^ = 0.01) in mean GH12 scores only according to the relationship status (married or cohabitating, engaged but not cohabitating, single), with Bonferroni post hoc tests showing a significative difference in GH12 scores (Table 3) between married cohabitating and engaged non-cohabitating (*p* = 0.004).

### 3.3. Impact of the Event (IES-R)

The overall mean IES-R score in participants was 22.90, with an SD of 15.27 (within a range from 0 to 88), reflecting a contingent moderate presence of PTSS.

The IES-R identified 76.5% of respondents with normal-level scores (score < 33) and 23.5% of respondents with problematic scores (score ≥ 33). IES-R scores were not normally distributed, so non-parametric analyses were considered.

The Kruskal Wallis-H test showed significative associations between IES-R and relationship status (χ^2^(2) = 27.74, *p* = 0.0001, the Mann-Whitney U test showing significative differences between married cohabitating and engaged non-cohabitating), and level of education (χ^2^(2) = 10.07 *p* = 0.006, the Mann-Whitney U test showing significative differences between low level of education and the other two levels).

The Mann-Whitney U test showed significative results for gender (U = 185,192.50, *p* = 0.0001), and maintenance of social network via the web (U = 138,206.50, *p* = 0.0001). Mean ranks are reported in Table 4.

### 3.4. Negative and Positive Emotions

Participants were asked to indicate which emotions they were feeling during the last week from a list of 13 emotions both negative and positive. Results (presented in Table 5) show that the most frequent emotion felt was sadness (72%), followed by boredom (54.5%), impotence (52%), and anxiety (50%). The least-felt emotions were: Blame (3.5%), disdain (8.6%), and surprise (10.5%).

### 3.5. Regression

Hierarchical regression models were run to identify the socio-demographic, daily life routine, and emotions factors impacting on participants’ GH12 global scores (Table 6). We ran preliminary Pearson’s correlations between the variables considered and inserted only the significative ones as independent variables in the regression models.

In the hierarchical regression models we entered the following independent variables: At the first step, socio-demographic factors; at the second one, the daily life routine factors and, at the last step, the presence/absence of emotions self-reported. The dependent variable was the GH12 score. In this regression the third model was the best one (R^2^ = 0.24; F = 28.68; *p* = 0.0001), with age, schooling years, working status, number of hours of work, maintenance of one’s social network via the web, and the presence of sadness, pain, interest, anger, surprise, happiness, boredom, impotence, anxiety, sense of guilt, and confusion impacting upon the General Health global score.

## 4. Discussion

To our knowledge, this study is one of the first to report the impact of the COVID-19 quarantine during April 2020 on the quality of life and symptoms of PTSD in a sample of the Italian adult population. The COVID-19 pandemic represents one of the most stressful recent events worldwide and poses a major challenge for the social, economic, and, above all, the psychological resources of the population, so it is very important to investigate also the psychological impact of this event.

The first finding of the present study was that the overall GH12 and IES-R scores in our participants indicated a contingent moderate stressful impact of COVID-19 on both QoL and PTSD symptoms, with percentages of problematic scores being very concordant, ranging from 23.5 (IES-R) to 24.5% (GH12) in the two instruments considered. These results are consistent with the findings reported by Lau et al. [3] and Zhang e Feei Ma [13], who investigated mental health and quality of life in Hong Kong residents during the SARS epidemic in 2003 and in Jinzhou in 2020, respectively, and found mild to high effects on the same parameters investigated here. The proportion of the population that feels stressed due to the pandemic is around one-fourth/one-third in many surveys [29].

The most frequent emotion felt during the quarantine in our study was sadness, followed by boredom, impotence, and anxiety. This result is quite consistent with those of previous research [5,29,30,31], even if among our participants the percentages of these negative emotions were much higher than in the precedent studies. Probably, Italian people suffered from the impossibility of meeting other persons, expecially whom they were fond of and missed the social relationships and social occasions, more than people in other countries did. In fact, it is typical in Italy to meet and go out very often with friends and the loved ones, expecially during weekends, and so having to give up this habit must have led the Italian people to experience much suffering through high rates of sadness and boredom. Obviously, among our participants there were also many concerns about the present and the future and hence a sense of helplessness/impotence and anxiety.

Regarding the impact of socio-demographic factors on the constructs measured here, being married or cohabitating, was associated with a higher quality of life and fewer PTSS, while people who were engaged but not cohabiting showed worse scores on both measures. The possibility to live together with a partner during the lockdown probably gave the chance to feel less worried and more secure in facing the quarantine, while those partners who lived separately were more worried for each other and suffered from the distance. With regards to relationships during the lockdown, Grover et al. [29] found that spousal/partner relationships improved even markedly in 47% of cases. In this sense the quarantine seemed to be an occasion to ameliorate the ménage in most cases, even if in other cases (those already problematic) the forced domestic partnership had the potential to wear a relationship out. The quarantine may have improved those relationships that were not at risk before the COVID-19 pandemic in different ways, providing availability of more free time to spend together with the partner, less work pressure, and possible fulfillment of long-desired sharing. Findings of the present study are supported also by many news stories in the media that reported positive aspects of the initial phase of the lockdown, such as making people realize the value of reconnecting with their family and improvement in love/dating and family relationships [29].

Results from the regression on the GH12 showed that people with higher education, people who were working during the quarantine, and people with a higher number of working hours declared worse QoL perceptions. The result about the association between higher education and worse well-being (more PTSS) emerged also from the non-parametric tests on the IES-R. It is probable that a higher schooling level provided a higher awareness of the dangers and risks linked to COVID-19, thereby worsening the perceived QoL and increasing PTSS in our sample. The literature is not consistent in this aspect. Research on another pandemic before COVID-19 identified a lower level of education as being associated with negative psychological impacts of the quarantine [4]. However, another study [32] suggested that demographic factors such age and education, were not associated with psychological outcomes. Workers and those spending more hours working reported more problems in their quality of life, probably due to the higher risk of being affected by COVID-19 if they needed to go out and, or, for those who kept working at home, due to a difficult adaptation to smart working, which was a big issue on the account of its novelty in Italy. A protective factor seemed to be the ability to maintain the social contacts via the web during the quarantine: People who expressed being able to maintain their social network using the media referred to better health quality of life and fewer PTSS. Staying at home for a long period of time limits individuals’ interaction with others whom they are used to interacting with often, making people suffer from isolation [33]. Using media to maintain contacts can alleviate this sense of loneliness and depression [34]. The WHO [35] recommended staying connected and maintaining social networks during the lockdown in order to maintain psychological well-being.

Results from the regression analyses on the GH12 with regards to emotions showed that those who expressed feeling more negative emotions also had a worse QoL. This result draws our attention to the fact that the presence of negative emotions is a key factor for a worse perception of quality of life and psychological well-being and, if soon identified, could be used as a screening measure to identify people at major risk of psychological suffering.

It will be useful in the future to compare these first data on the initial phases of the quarantine with successive follow-ups. Although the psychological impact is greater during the quarantine, it is plausible that there are also long-term consequences. Comparing the psychological outcomes during the quarantine for SARS with subsequent psychological outcomes, Jeong et al. [5] showed that during the quarantine, 7% of subjects reported anxiety, and 17% anger; 4–6 months after the end of the quarantine, these percentages had decreased, although anxiety (3%) and anger (6%) were still present. Other long-term effects associated with the SARS quarantine—namely, alcohol abuse and addiction symptoms—emerged from a study conducted by Yoon et al. [36] on health workers 3 years after the SARS epidemic. Moreover, even a few months after the end of the quarantine period, people continued to avoid closed and/or crowded places or coughing people [6] and adopted control behaviors such as careful hand washing [9]. Therefore, many people reported the difficulty of returning to normal for a long time after the end of the lockdown, and this aspect deserves to be investigated also in relation to the COVID-19 lockdown.

### Strengths and Limitations

Our study has some strengths and limitations. To the best of our knowledge, it is one of the first studies to provide an opportunity to investigate the impact of the COVID-19 pandemic, being conducted 2 weeks after the lockdown imposed by the Italian government. This is particularly important, as this study presents some of the first data about the quality of life and PTSS impacts of the COVID-19 lockdown. However, our study has some limitations associated with the convenience sampling method, which limits the generalization of our findings. Moreover, the temporal window of data collection must be considered as a limit, as it started 2 weeks after the beginning of the lockdown and lasted 1 month. In this limited period, people probably did not completely realize the potential consequences of the pandemic and quarantines. We cannot foresee permanent changes in the quality of life after 2 weeks after the occurrence of a potentially aversive stimulus such as the pandemic and its consequent quarantine. We can rather expect increased levels of stress or anxiety and, consequently, a gradual deterioration of the quality of life in the successive phases. We must be aware that these data are just a snapshot of a contingent situation. We, of course, do not have data about the quality of life and PTSS of the participants before the quarantine, so we cannot be sure on whether their previous conditions were better or worse. We cannot interpret the present results as a change due to the lockdown but must consider them just as a contingent description of such a difficult moment. Another limitation is that without a strict comparison between lockdown and non-lockdown areas, it is hard to distinguish the effect of the quarantine from that of the COVID-19 pandemic. To date, there is no literature on this distinction to refer to. Moreover, it is possible that our participants were limited mostly to those who had financial, emotional, and social support to answer our questions, which might have skewed all of the answers towards those expected from a “healthier” population in this context. Furthermore, due to the online questionnaire being a self-report evaluation, the indicated levels of health quality of life and PTSS might not consist of those with an evaluation by mental health professionals.

## 5. Conclusions

The main result of this study was that the COVID-19 quarantine during April 2020 was associated with a moderate-to-severe stressful impact in our sample. Since the COVID-19 pandemic is still ongoing, these findings need to be confirmed and investigated in future follow-ups.

Our study managed to capture some immediate impacts of the COVID-19 lockdown in Italy, a country that was badly affected by the virus. The quarantine measure was certainly essential in the fight against the spread of COVID-19, but in order to truly pay attention to the health of citizens, it is vital—in addition to strengthening medical health activities—to take into consideration the effects that this circumstance has generated in people. The stress experienced was related to not only the presence of the virus but also the measures chosen to counter it. Regardless of the need or lack of need to institute a lockdown, it is advisable that institutions during the decision-making phase take into consideration the psychological costs that such a measure implies, striving to reduce as much as possible the negative consequences associated with it. It is fundamental to understand how quarantining during a lockdown can impact on adults’ quality of life and PTSS presence and which main daily and socio-demographic parameters can be associated with it.

The energies that will be dedicated from now on to mental health must be increased, since we are facing, in accordance with reporting by the Italian National Council of Psychologists, not only a health but also a psychological emergency. This term refers to the possible development in a massive way of symptoms and disorders related to anxiety and trauma that were experienced during the pandemic. The COVID-19 pandemic, totally unexpected and sudden, caught everyone completely unprepared, and the consequences associated with it must be evaluated and taken into consideration.

## Figures and Tables

**Table 1 ijerph-18-04385-t001:** Participants’ characteristics.

Variable	Characteristic	Frequency	%
Gender	Males	409	22.2
	Females	1430	77.8
Level of education *	Low	134	7.3
	Medium	854	46.4
	High	851	46.3
Occupation status	Working	1068	58.1
	Not working	770	41.9
Relationship status	Married/Cohabitating	642	34.98
	Engaged, not cohabitating	613	33.3
	Single	584	31.8
	Range	Mean	SD
Males’ age	18–73	33.33	13.84
Females‘ age	18–72	32.42	12.79
Number of smart-working hours	0–16	3.43	3.44
Number of people living together during quarantine	0–15	2.38	1.44

* Level of education classification: Low = ISCED 0 to 2, primary and middle school; Medium = ISCED 3 to 5, upper secondary school; High = ISCED 6 to 8, bachelor, master, PhD diploma.

**Table 2 ijerph-18-04385-t002:** Descriptive statistics of daily life parameters.

Daily Life Parameter	N (%)
Precedent pathologies	No: 1649 (89.7) Yes: 190 (10.3)
Separation from people whom I used to live with before the quarantine	No: 1486 (80.8) Yes: 353 (19.2)
Familiars directly involved in the emergency	No: 1530 (82.9) Yes: 316 (17.1)
Economic difficulties caused by COVID-19	No: 1369 (74.4) Yes: 47 (25.6)
Economic difficulties not caused by COVID-19	No: 1642 (89.3) Yes: 197 (10.7)
Presence of close persons affected by COVID-19 with severe symptoms	No: 1792 (97.1) Yes: 54 (2.9)
Familiars with a disability	No: 1571 (85.4) Yes: 268 (14.6)

**Table 3 ijerph-18-04385-t003:** GH12 scores (mean and SD) according to the relationship status.

Relationship Status	Mean	SD
Married	17.26	4.94
Engaged	18.33	6.07
Single	18.01	5.85

**Table 4 ijerph-18-04385-t004:** IES-R mean ranks according to the independent variables.

Variable	Level of the Variable	N	Mean Rank
Relationship status	Married	642	850.17
	Engaged	613	1006.39
	Single	584	906.08
Level of education	Low	134	783.18
	Medium	854	921.94
	High	851	939.60
Gender	Male	409	657.69
	Female	1430	994.99
Maintenance of social network via web	Yes	1775	814.80
	No	64	1064.14

**Table 5 ijerph-18-04385-t005:** Descriptive analyses of emotions felt by participants in the last week.

Emotion	Frequency	Percentage
Sadness	1324	72
Boredom	1002	54.5
Impotence	956	52
Anxiety	920	50
Happiness	709	38.6
Anger	686	37.8
Confusion	642	34.9
Interest	560	30.5
Pain	287	15.6
Sense of guilt	221	12
Surprise	193	10.5
Disdain	158	8.6
Blame	65	3.5

**Table 6 ijerph-18-04385-t006:** Hierarchical regression analysis on GH12 scores.

Step	Variables	*R^2^*	Δ*R^2^*	*F*	*p*	*β*	*p*	*VIF*
1	Socio-demographic factors	0.02	0.02	12.54	0.0001			
2	Daily life factors	0.05	0.02	13.81	0.0001			
3	Emotions (0 = no; 1 = yes)	0.24	0.19	29.68	0.0001			
	Age					0.074	0.04	1.05
	Gender					−0.011	ns	1.12
	Schooling years					0.043	0.04	1.09
	Working/Not working					−0.96	0.0001	1.17
	Number of daily hours of work/study					0.07	0.001	1.06
	Separation from people during the quarantine					0.025	ns	1.02
	Maintenance of social network via the web					−0.1	0.0001	1.12
	Sadness					0.08	0.0001	1.13
	Pain					0.07	0.001	1.08
	Interest					−0.12	0.0001	1.13
	Anger					0.05	0.01	1.13
	Surprise					−0.068	0.001	1.08
	Happiness					−0.16	0.0001	1.12
	Blame					0.03	ns	1.09
	Disdain					0.006	ns	1.11
	Boredom					0.1	0.0001	1.06
	Impotence					0.083	0.0001	1.1
	Anxiety					0.183	0.0001	1.08
	Sense of guilt					0.11	0.0001	1.08
	Confusion					0.1	0.0001	1.03

## Data Availability

The data presented in this study are available on request from the corresponding author.

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
