# Peer review of "Quality of Life and Symptoms of PTSD during the COVID-19 Lockdown in Italy"

_ijerph, 2021, doi:10.3390/ijerph18084385_

Round 1

Reviewer 1 Report

The improvements are correct and respond to the suggested requirements. However, concerning the results, more connections across subsections will be needed to make clearer the synergies across results and improve the scientific soundness of the contribution. 

Author Response

Thank you for your suggestion, we tried to add more connections in the results section to make clearer the synergies across results

Reviewer 2 Report

Thank you very much for the opportunity to review this manuscript. It is of great significance to investigate the psychological consequences of the lockdown policy. I have some comments to help improve the quality of this maunucript:

  1. As the authors stated, there have been rich literature concerning the impact of COVID-19 quarantine on quality of life and symptoms of PTSD. The author highlighted timing “initial phase of the COVID-19 lockdown” as their contribution to knowledge. As the first research aim stated “How did Italian adults report their health related QoL and PTSS during April 2020, after two weeks after the COVID-19 quarantine started…”. I assumed that the “initial phase” referred to “2 weeks’ time”, and you were arguing that one’s feeling/emotions is time-sensitive. However, the cross-sectional study was performed between 1 April and 30 April, which indicated that the last respondent submitted his/her answer after 6 weeks after the COVID-19 quarantine started. Can this response reflect one’s feeling/emotions at the initial phase of the COVID-19 lockdown? Please justify this.
  2. Please remove Tables 1 & 2 to the results section, as it may be difficult for readers to understand the meaning of “level of education” before you introducing research instruments. Regarding research instruments, please introduce the classification of some variables such as level of education, qualification achieved.
  3. Please provide a descriptive analysis of the sociodemographic conditions of your respondents. I did not find information regarding qualification achieved, type of housing, economic status, place of residence, etc.
  4. The authors stated in the statistical analysis plan section (2.4) that they employed ANOVA to assess the difference in dependent variables according to relationship status. Please justify why only relationship but not other sociodemographic variables?
  5. Table 4, it seems that IES-R was not normally distributed. In this case, it may not be appropriate to use Mean SD to describe it.
  6. Tables 6 & 7, please provide VIF value to check the collinearity issue, as some variables seemed to be associated with each other, for instance, happiness and sadness.
  7. Table 6 indicated the association between schooling years and GH12 as β=0.043. However, the authors stated in the discussion section “people with higher education.. declared worse QoL perceptions.”. They seemed to be contradictory.
  8. The authors stated in the limitation section that “We can rather expect increased levels of stress or anxiety and, consequently, a gradual deterioration of the quality of life in the successive phase.” Please justify this statement, as prior studies have witnessed a flattened patterns as one got used to lockdown, and perceived larger psychological distance with infected persons.

Author Response

As the authors stated, there have been rich literature concerning the impact of COVID-19 quarantine on quality of life and symptoms of PTSD. The author highlighted timing “initial phase of the COVID-19 lockdown” as their contribution to knowledge. As the first research aim stated “How did Italian adults report their health related QoL and PTSS during April 2020, after two weeks after the COVID-19 quarantine started…”. I assumed that the “initial phase” referred to “2 weeks’ time”, and you were arguing that one’s feeling/emotions is time-sensitive. However, the cross-sectional study was performed between 1 April and 30 April, which indicated that the last respondent submitted his/her answer after 6 weeks after the COVID-19 quarantine started. Can this response reflect one’s feeling/emotions at the initial phase of the COVID-19 lockdown? Please justify this.

The respondents submitted their answers starting after two weeks since the beginning of the quarantine, during April 2020 (so from the third to the sixth week from the beginning), so we can consider this period the initial phase of the lockdown, as restrictions are lasting also nowadays. As many weeks of lockdown followed April 2020, we can consider that period an initial phase in this sense.

Please remove Tables 1 & 2 to the results section, as it may be difficult for readers to understand the meaning of “level of education” before you introducing research instruments. Regarding research instruments, please introduce the classification of some variables such as level of education, qualification achieved.

We moved Table  2 to the results section and described it. We operationalized the classification of level of education in the Instruments’ section as requested and added a legend in Table 1, so that the reader can understand the levels also before reading the Instrument section.

Please provide a descriptive analysis of the sociodemographic conditions of your respondents. I did not find information regarding qualification achieved, type of housing, economic status, place of residence, etc.

As we did not find any significative results about the sociodemographic conditions of our respondents we mentioned in the Instrument section (qualification achieved, type of housing, economic status, place of residence), we avoided mentioning them and removed these variables from this section.

The authors stated in the statistical analysis plan section (2.4) that they employed ANOVA to assess the difference in dependent variables according to relationship status. Please justify why only relationship but not other sociodemographic variables?

We run analyses also on other sociodemographic variables, but did not find any significative result, that’s why we reported results only according to relationship status. We justified it in the text.

Table 4, it seems that IES-R was not normally distributed. In this case, it may not be appropriate to use Mean SD to describe it.

As a matter of fact IES-R was not normally distributed, so we run non-parametric analyses. We left the first information about mean and standard deviation as it is useful to interpret our sample scores within the IES-R normative parameters. We added all the non-parametric analyses referring to independent variables and eliminated the regression on IES-R (Table 7).

Tables 6 & 7, please provide VIF value to check the collinearity issue, as some variables seemed to be associated with each other, for instance, happiness and sadness.

We provided VIF value to check the collinearity issue in Table 6 and luckily we do not have any problem of collinearity.

Table 6 indicated the association between schooling years and GH12 as β=0.043. However, the authors stated in the discussion section “people with higher education.. declared worse QoL perceptions.”. They seemed to be contradictory.

This is correct in the discussion, as higher scores in GH12 mean worse quality of life

The authors stated in the limitation section that “We can rather expect increased levels of stress or anxiety and, consequently, a gradual deterioration of the quality of life in the successive phase.” Please justify this statement, as prior studies have witnessed a flattened patterns as one got used to lockdown, and perceived larger psychological distance with infected persons.

In Italy we are facing an exacerbation of stress level as freedom is so limited and lockdown is producing a lot of economic problems and logistic problems in families with children attending school online remaining at home. That’s why we stated that increased levels of stress or anxiety is expected in a successive phase.

Reviewer 3 Report

Dear Authors,

It's been a long time since I've seen such a great improvement in a paper between edits. The authors deserve to be congratulated, because from a paper with a very low scientific level the work now presents a high scientific level. I am only puzzled by the additional information included in the analysis, but apparently the authors wanted to divide the material into several papers. I am glad that these new elements completed the manuscript because it is now coherent. I assume that an additional person supported the authors' efforts but was not listed as a co-author. Substantively, I have no complaints, but the editorial and graphic layout is abysmal. It seems to me, however, that adopting a "clean version" will make the manuscript easier to follow. 

Author Response

Thank you for your positive comments. We will work on the editorial and graphic layout with the help of the editorial board, in order to have an optimal version of the manuscript easy for readers to follow. English language spell check has been provided.